# Wa-1 Equine-Like G3P[8] Rotavirus from a Child with Diarrhea in Colombia

**DOI:** 10.3390/v13061075

**Published:** 2021-06-04

**Authors:** Marlen Martinez-Gutierrez, Estiven Hernandez-Mira, Santiago Rendon-Marin, Julian Ruiz-Saenz

**Affiliations:** 1Grupo de Investigación en Ciencias Animales GRICA, Universidad Cooperativa de Colombia, Bucaramanga 680002, Colombia; marlen.martinezg@campusucc.edu.co (M.M.-G.); estiven.hernandezm@udea.edu.co (E.H.-M.); santiago.rendonm@udea.edu.co (S.R.-M.); 2Infettare, Facultad de Medicina, Universidad Cooperativa de Colombia, Medellín 050012, Colombia

**Keywords:** rotavirus A, diarrheal disease, emerging virus, interspecies surveillance and reassorting strains

## Abstract

Rotavirus A (RVA) has been considered the main cause of diarrheal disease in children under five years in emergency services in both developed and developing countries. RVA belongs to the *Reoviridae* family, which comprises 11 segments of double-stranded RNA (dsRNA) as a genomic constellation that encodes for six structural and five to six nonstructural proteins. RVA has been classified in a binary system with Gx[Px] based on the spike protein (VP4) and the major outer capsid glycoprotein (VP7), respectively. The emerging equine-like G3P[8] DS-1-like strains reported worldwide in humans have arisen an important concern. Here, we carry out the complete genome characterization of a previously reported G3P[8] strain in order to recognize the genetic diversity of RVA circulating among infants in Colombia. A near-full genome phylogenetic analysis was done, confirming the presence of the novel equine-like G3P[8] with a Wa-like backbone for the first time in Colombia. This study demonstrated the importance of surveillance of emerging viruses in the Colombian population; furthermore, additional studies must focus on the understanding of the spread and transmission dynamic of this important RVA strain in different areas of the country.

## 1. Introduction

Diarrhea is one of the most common diseases in infants and young children in both developed and developing countries. The incidence of diarrhea in African, Latin American, and Asian countries in children under 5 years of age has been estimated to be over one billion, with approximately 3.3 million deaths per year [1]. After the introduction of rotavirus vaccine in more than 60 countries worldwide, the number of rotavirus A-associated deaths in children under 5 years of age declined from 528,000 (range, 465,000–591,000) in 2000 to 215,000 (range, 197,000–233,000) in 2013 [2] and to 128,500 (range 104,500–155,600) throughout the world in 2016 [3].

Before the introduction of rotavirus vaccine in Latin America, more than 15,000 deaths annually were related to Rotavirus A (RVA) infection; Latin American countries have shown a decrease in mortality by diarrhea after rotavirus vaccine was implemented [2,4]. Specifically in Bucaramanga City (the north-eastern region of Colombia), prior to the vaccine introduction, RVA was considered the main cause of diarrheal disease in children under five years who consulted emergency services, accounting for 44.4% (95% confidence interval [CI] 37.1–52.0) of the reported cases [5]. Recent data showed that the prevalence of rotavirus infections had decreased to 30.53% (95% [CI] = 21.2–39.7%) [6].

RVA is a member of the *Reoviridae* family, comprising 11 segments of double-stranded RNA (dsRNA) as genomic constellation that encode six structural proteins (VP1–VP4, VP6 and VP7) and five nonstructural proteins (NSP1–NSP5/6). RVA is classified in a binary system as Gx[Px] based on the spike protein (VP4) and the major outer capsid glycoprotein (VP7). In addition, whole-genome classification is used to assign genotypes to each gene (Gx-P[x]-Ix-Rx-Cx-Mx-Ax-Nx-Tx-Ex-Hx) [7,8].

Most human RVA genomes have been assigned to three genotype constellations: Wa-like or genogroup 1 (G1/3/4/9/12 P[8]-I1-R1-C1-M1-A1-N1-T1-E1-H1), DS-1-like or genogroup 2 (G2-P[4 ]-I2-R2-C2-M2-A2-N2-T2-E2-H2) and the less common AU-1-like or genogroup 3 (G3-P[9]-I3-R3-C3-M3-A3-N3-T3-E3-H3) [8]. Recently, the emergence of novel human inter-genogroup reassorting strains, DS-1-like G1P[8] and G3P[8], has been identified worldwide including Asia, Australia, Europe and America [9,10,11,12,13,14]. Moreover, atypical DS-1-like G3P[8] strains resulted from a rare human/equine reassortment event, named equine-like G3P[8] DS-1-like strains [13], have been widely distributed in South America, mainly in Brazil [12,15].

Currently, there are at least 36 G-genotypes and 51 P-genotypes reported worldwide, and at least 80 combinations have been identified in the RVA strains that infect humans and animals [16]. Globally, the majority of human RVA have the genotype combination of G1P[8], G2P[4], G3P[8] and G4P[8]; however, a high prevalence of the so-called emerging strains G9P[8] and G12P[8] strains has been reported [17]. In Colombia, the most common G[P] genotype combinations in recent studies are G3P[8] followed by G3P[9] [6]; however, to date, no full constellation of any of those viruses has been reported yet.

Multiple factors have been discussed that could trigger the RVA evolution. Interspecies transmission and reassortment between human and animal RVA strains strongly contribute to the evolution of RVA [18,19]. In addition, the introduction of RVA vaccines into human populations may increase serotype replacement into non-vaccine serotypes over time, due to a potential vaccine selective pressure, possibly leading the evolutionary rate and the capability of new RVA strains to diffuse in human populations [20,21].

Due to the strong influence of G3 strains in recent years in Colombia, the global increase of Equine-like patterns in G3 strains in the Americas and the lack of full genome sequencing of RVAs in Colombia, the aim of the present study was to conduct complete genome characterization of a previously reported G3P[8] strain in order to recognize the genetic diversity of RVA circulating among infants in Colombia.

## 2. Materials and Methods

### 2.1. Type of Study and Samples

This was a retrospective and descriptive study conducted in a G3P[8] positive sample belonging to a cohort study previously analyzed [6]. Sample (RVA/Human-wt/COL-BUC/55/2015/G3P[8]) belonged to a two-year-old child who presented with acute gastroenteritis at a public hospital belonging to the Bucaramanga Institute of Health—ISABU in the north-eastern region of Colombia in November 2015.

### 2.2. RNA Extraction

Stool suspensions (10%) were prepared in 0.01 M phosphate-buffered saline (PBS) (pH 7.2), vortexed and centrifuged. The supernatant was filtered using a 0.22-µm filter and transferred to a sterile centrifuge tube. The total RNA was extracted from 140 µL of the fecal suspension using the QIAamp Viral RNA extraction kit (QIAGEN, Hilden, Germany) according to the manufacturer’s instructions. The RNA quality and quantity were determined spectrophotometrically by using a NanoDrop ONE™ spectrophotometer (Thermo Fisher Scientific Inc., Waltham, MA, USA) with 1 μL of sample. RNA aliquots were stored at −70 °C until used for PCR.

### 2.3. Amplification of Equine-Like G3P[8] Genome

One-step RT-PCR was carried out using the SuperScript^®^ III One-Step RT-PCR System with Platinum^®^ Taq DNA Polymerase. (Life Technologies™, Paisley, UK). Each reaction was made by adding 12.5 μL 2× reaction mix, 1 μL enzyme mix, 1 μL primers (10 μM), 0.5 μL RNA extract and nuclease-free water. The amplification conditions included 60 °C for 30 min, 94 °C for 2 min followed by 45 cycles of 94 °C for 15 s, 55 °C for 30 s, 68 °C for 2 min and a final extension of 68 °C for 5 min. Previously reported primers [22] targeting the conserved 5′- and 3′-end regions as well as internal primers were used for RT-PCR and sequence were used (Appendix A).

PCR amplification results were visualized using 1.5% horizontal agarose gel electrophoresis. Gels were stained with the Invitrogen ™ SYBR^®^ Safe DNA Gel Stain (Thermo Fisher Scientific^®^). In each well, 4.2 μL of each sample obtained after amplification and 0.8 μL of the 6× DNA loading buffer were used, and the GeneRuler™ 100-bp DNA Plus Ladder (Thermo Fisher Scientific^®^) was used as a molecular weight marker. Gels were developed in the ultraviolet light Gel Doc™ XR+ imaging system (Bio-Rad, Molecular imager^®^, Hercules, CA, USA) and were visualized using ImageLab™ software. The amplicons of RT-PCR were purified and sequenced using SANGER technology by Macrogen Inc. (Seoul, Korea) in an automated sequencer. The sequences obtained were compared with various prototype strains of each genotype by using BLAST™ and the RotaC v2.0 automated genotyping tool for Group A rotaviruses [20] in the Virus Pathogen Resource (ViPR) online. Sequences were deposited with GenBank accession numbers MZ209037 to MZ209044.

### 2.4. Phylogenetic Analyses

The reported electropherograms from the sequencing were analyzed using the Chromas™ v. 2.6 software. Contig generation, resulting from the overlapping of the sequences amplified by the primers, was performed on the SeqMan Pro platform with the Lasergene™ tool. After constructing the complete nucleotide sequences for each sample, alignment was performed using the ClustalW method, after which these sequences were compared with the cDNA sequences obtained from GenBank. All analyses were performed in the MEGA™ 7.0 software for Windows^®^. For the phylogenetic analysis, we calculated the best nucleotide substitution model for the dataset generated with the sequences from each gene Gx-P[x]-Ix-Rx-Cx-Mx-Ax-Nx-Tx-Ex-Hx obtained from GenBank. A maximum-likelihood tree was constructed for each genome segment. The best substitution models were selected based on the corrected Akaike Information Criterion (AICc). The models used in this study were Tamura 3-parameter (T92) +G (NSP3, NSP4, VP3, VP4, VP6 and VP7), T92 +I (NSP2, VP1) and Tamura-Nei (TN93) +G (VP2). A bootstrap value of 1000 was used in all cases. All phylogenetic analyses were developed in MEGA™ 7.0 software for Windows^®^.

## 3. Results

By using the RotaC v2.0 web-based classification tool, the Colombian RVA/Human-wt/COL-BUC/55/2015/G3P[8] strain had a Wa-like backbone and a genotype constellation as G3-P[8]-I1-R1-C1-M1-Ax-N1-T1-E1-Hx. We were unable to amplify the NSP1 and NSP5 genes. At BLAST analysis (Table 1), we found that most sequences of the constellation had high identity (>99.4%) to human G12P [8] strains from Spain (RVA/Human-wt/ESP/SS62505693/2013/G12P [8]) and Italy (Hu-wt/ITA/PA93/12/2012/G12P [8]).

Interestingly, most of the evaluated genes (P[8]-R1-C1-M1-I1-N1-T1-E1) display a high identity to Bovine RVA previously reported in Uganda (RVA/Cow-wt/UGA/BUW-14-A035/2014/G12P[8]), which has been highly related to human strains. On the other hand, at BLAST, G3 displays the highest identity (99.69%) to Brazilian Equine-Like G3 reported in 2015 (RVA/Human-wt/BRA/IAL-R594/2015/G3P[8]—RVA/Human-wt/BRA/IAL-R330/2015/G3P[8]).

The phylogenetic analysis of the different genes confirmed that the equine-like G3 strain from Colombia clearly clustered with other Equine-Like strains (Figure 1) that have been reported in other countries, in a paraphyletic group to Equine RVA strains.

For comparison purposes, all genes (I1-R1-C1-M1-N1-T1-E1-H1) were analyzed with the same dataset. As can be seen in Figure 2, nucleic acid sequences coding for structural and non-structural proteins clustered (Bootstrap >99%) to Wa-like prototype strains in genotype 1 (Table 1—RVA/Human-tc/USA/Wa-20-AG/1974/G1P[8]) at completely different branches from the Equine-like strains sequences.

## 4. Discussion

Here, we describe the presence of the emerging equine-like G3P[8] rotaviruses in Colombia. However, the present report also shows the intergenogroup reassortment between an Equine-like G3P[8] and a “classical” Wa-like backbone strain. Furthermore, it is well known that almost all the reports of equine-like G3P[8] human infections have been associated to a DS-1-like genogroup (I2-R2-C2-M2-A2-N2-T2-E2-H2) [11,14,15].

Although a robust bias has been reported towards the maintenance of pure Wa-like genome constellations for the G3P[8] isolates [23], RVA reassortment between different strains has been recognized as one of the most important mechanisms governing the evolution of RVA [12]. Furthermore, reassortment may give rise to interspecies transmission with subsequent adaptation to the human host for the new viral variant [24]. Equine-like G3P[8] DS-1-like strains first emerged in Asia and Europe between 2013 and 2015 and they have since been distributed in many places around the world, including the Americas, which shows that the novel G3 strains have a high potential of rapid worldwide spread [9,11,13,14,15].

As has been previously reported, the equine-like G3 strains that was found in Colombia exhibited a high nucleotide identity through the whole genome of the equine-like G3 strains distributed worldwide [25,26]. Moreover, as was recently described by Akane et al., in comparison with the Japanese Equine-like G3 strain, the Colombian VP7 genes from the equine-like G3 had low nucleotide identity with those of equine/dog G3 strains (77–84%), as well as lower identity with other common human G3 strains present on the NCBI data base (55.4% to the Wa-like RVA/Human-tc/USA/Wa/1974/G1P8 (JX406755)) [25].

Multiple reports showing the emergence of Equine-like G3P[8] strains confirm that these strains have genetically evolved through reassortment events between Wa and DS-1 genogroups [11,13]. Furthermore, it has been recently suggested that the equine-like G3 strain is a mono-reassorting strain of the VP7 gene (unknown animal origin) and the other 10 genes from DS-1-like G1P[8] strains [25]. The increasing concern is the fact that explains why the DS-1-like G3P[8] strains have been detected in different geographical areas in association with severe childhood rotaviral gastroenteritis [13], even in school-aged children without any known severe underlying problems, showing the potentially high severity of those emerging strains [27].

The backbone of the Colombian RVA was Wa-like with VP7 and VP4 genes from an Equine-like G3 origin (Figure 1). Our results not only confirm the circulation in Colombia of Equine-like strains, but also emphasize the need of surveilling the circulation of DS-1-like strains with and without Equine G3 VP7 genes, as they have been commonly reported in Brazil [15,28] and the Dominican Republic, with further evidence of reassortment of the DS-1 equine-like G3 strains with locally-circulating strains [29]. We believe that the emergence of Wa-1 equine-like G3P[8] in this study may be the result of a rearrangement between a non-previously reported equine-like-DS-1 variants and a local Wa-like strain as has been reported elsewhere [29]. Since G12P [8] strains have been recently reported in Colombia [6], it could be possible that backbone strains belong to local G12P [8] with a high identity (>99.4%) to human G12P [8] strains from Spain (Table 1).

An important weakness of our analysis was the inability to amplify and analyze NSP1 and NSP5 coding genes. NSP1 sequence comparison from different RVA had revealed that the NSP1 coding gene is highly variable in comparison to other RVA genes [30]. The NSP5 gene has been the most frequently viral segment involved in RVA genome rearrangements in all animal RVA strains [31]. The lack of those viral genes adds a strong bias to the complete genome analysis of the RVA herein reported, highlighting the importance of carrying out a full/whole genome analysis in order to accomplish active surveillance and characterization of circulating RVA strains in the post-vaccine introduction, besides the identification of the evolutionary process, which has led to the emergence of atypical strains under vaccine pressure, as has been suggested elsewhere [32]. Additionally, it has been observed that equine-like G3 may be mistyped in conventional genotyping approaches because of lower primer specificity [13,14], underestimating the reports and prevalence of this RVA emerging strain in Colombia. The high genetic variability of the NSP1 and NSP5 genes [16], lower primer efficiency or even low RNA quality due to the long storage time of patient stool samples [33] may help explain the difficulties in amplifying the missing segments, leading to difficulties for the knowledge of the genomic constellation of the RVA from this patient.

The immune pressure associated with RVA vaccines has been reported as the main driving force for the emergence and reassortment of new animal/human RVA strains [29,32]. These facts, besides being attributed to the spread of Equine-like G3 strains, have been attributed to Rotarix^®^-induced selective pressure. Remarkably, a recent study including analysis of two decades of Australian surveillance data found that G1P[8] strains dominated the pre-vaccine era (1995–2006), while following vaccine introduction (2007–2015), genotype distribution varied based on localized vaccine use, with G12P[8] strains dominant in areas using RotaTeq^®^, and Equine-Like G3P[8] and G2P[4] strains dominant in areas using Rotarix^®^ [34]. Globally, there has been a relative decrease in Wa-like G1P[8] and a relative increase in G2P[4], with a traditional DS-1-like constellation; moreover, RVA genotype diversity has increased worldwide [25]. Although, those changes could be attributed to the natural fluctuation of endemic/epidemic strains, the selective pressure of the RVA vaccine is still a big concern and studies on different epidemiological panoramas must be carried out.

In Latin America, especially in Colombia, despite Rotarix^®^ vaccination, RVA diarrhea is still a main concern among infants and young children. A recently published case-control study assessing the etiology of moderate to severe acute gastroenteritis in children that were less than 5 years of age, carried out in the same city from the patient of our current study (Bucaramanga, Colombia), found that norovirus and rotavirus explained the major proportion of moderate to severe clinical cases [35]. However, after the introduction of RVA vaccination in 2009, there was a reduction of 39% in diarrhea-related mortality in children <5 years old (95% CI, 35 to 44) in Colombia [36], with a rapid switch from Wa-like to DS-1-like strains [37] and a high frequency of unusual G9P[4] in 2012 [38] and G3 P[8]/P[9] strains accounting for the vast majority of cases (82.8%) in 2015–2016 [6]. As has been seen in other countries in Latin America, the strong bias towards G3 strains, mainly in Brazil, is mostly due to the presence of the Equine-like RVA strains [12,15]. This data led us to propose that a wider molecular analysis must be carried out in Colombia to better understand the current implication of the Equine-like RVA strains in the epidemiology of RVA in Colombia.

It is noteworthy that most of the genes from the present G3P[8] strain circulating in Colombia are phylogenetically related to a human/bovine strains previously reported in Uganda [39], which has been proposed as being the result of human to animal RVA transmission due to the bovine virus gene sequences sharing high identities at the nucleotide level to those of the human RVA strains belonging to the same lineages previously reported [14,40]. Thus, these facts highlight the importance of the interspecies surveillance to fully understand the molecular epidemiology of the RVA and the full impact of the interspecies transmission in the RVA associated disease in the post-vaccine era around the world.

## 5. Conclusions

This report describes for the first time in Colombia the presence of the novel equine-like G3P[8] with a Wa-like backbone, suggesting that this important worldwide emerging viral variant must be included in routine surveillance of RVA in Colombia.

## Figures and Tables

**Figure 1 viruses-13-01075-f001:**
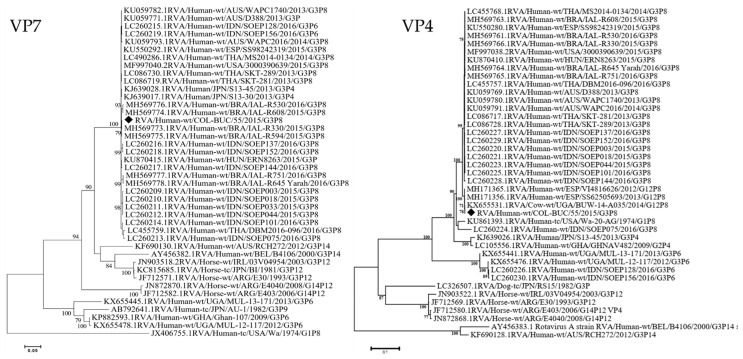
RVA VP7 (G)/VP4 [P] Maximum Likelihood Phylogeny. The trees with the highest log likelihood are shown. The percentage of trees in which the associated taxa clustered together is shown next to the branches. The tree is drawn to scale, with branch lengths measured in the number of substitutions per site. All positions containing gaps and missing data were eliminated. Bootstrap values lower than 70% are not shown. Black Diamonds (♦) indicate the Colombian G3P[8] strain in this study compared with other representative human and animal strains from the GenBank database. Evolutionary analyses were conducted in MEGA7.

**Figure 2 viruses-13-01075-f002:**
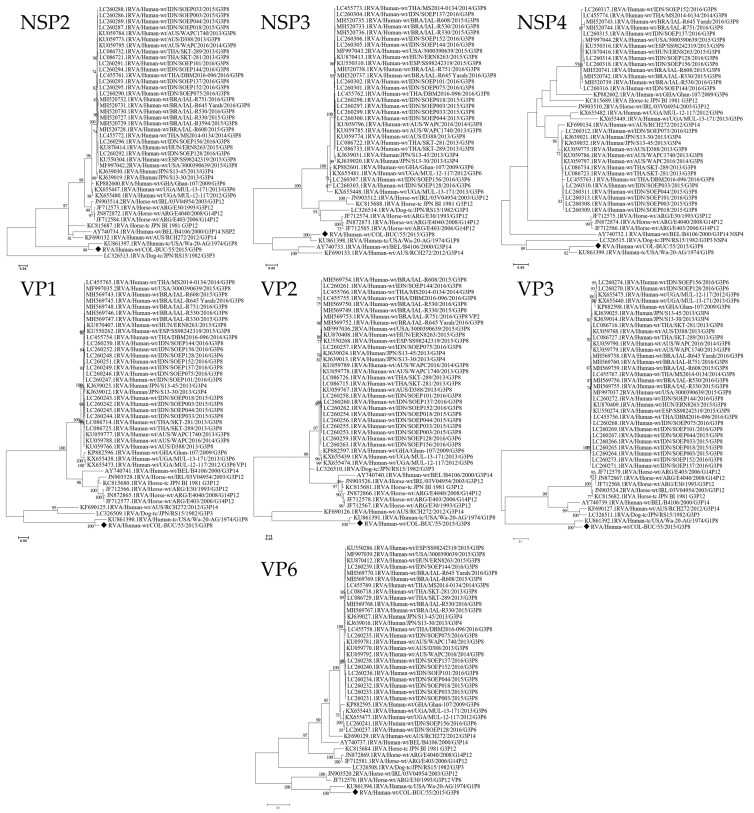
Maximum Likelihood Phylogeny of RVA encoding genome segments of G3P[8] rotaviruses possessing a Wa-like genetic backbone (I1-R1-C1-M1 -N1-T1-E1-H1), compared with other representative human and animal strains from the GenBank database. Bootstrap values lower than 70% are not shown. Scale bars indicate the number of substitutions per nucleotide position. Black Diamonds (♦) indicate the Colombian G3P[8] strain in this study. Evolutionary analyses were conducted in MEGA7.

**Table 1 viruses-13-01075-t001:** Prototype strains of each genotype by using BLAST™ and the RotaC v2.0 in ViPR.

Gen	Type	Viral Segment	Closest Strain	Covery	Ident%	GenBank
VP1	R1	1	RVA/Human-wt/BGD/Dhaka25/2002/G12P8	100	98.29	viprbrc.org
			RVA/Human-wt/ESP/VI4816626/2012/G12P [8]	100	99.5	MH171317
			RVA/Human-wt/ESP/SS62505693/2013/G12P [8]	100	99.5	MH171308
			RVA/Cow-wt/UGA/BUW-14-A035/2014/G12P [8] *	100	99.5	KX655528
			RVA/Human-tc/USA/Wa-20-AG/1974/G1P [8] ***	100	99.9	KU861390
VP2	C1	2	RVA/Human-wt/BGD/Dhaka12/2003/G12P6	100	99.19	viprbrc.org
			RVA/Human-wt/ESP/SS66209011/2013/G12P [8]	100	99.9	MH171327
			RVA/Human-wt/ESP/SS257451/2012/G12P [8]	100	99.9	MH171322
			RVA/Cow-wt/UGA/BUW-14-A035/2014/G12P [8] *	100	99.9	KX655529
			RVA/Human-tc/USA/Wa-20-AG/1974/G1P [8] ***	100	99.9	KU861391
VP3	M1	3	RVA/Human-wt/BGD/Dhaka12/2003/G12P6	99.72	98.54	viprbrc.org
			RVA/Human-wt/ESP/VI4816626/2012/G12P [8]	100	99.5	MH171349
			RVA/Human-wt/ITA/PA144/12/2012/G12P8	100	99.5	KU048574
			RVA/Human-wt/ITA/PA93/12/2012/G12P8	100	99.5	KU048571
			RVA/Human-tc/USA/Wa-20-AG/1974/G1P [8] ***	100	99.9	KU861392
VP4	P[8]	4	RVA/Human-wt/BGD/Dhaka25/2002/G12P8	100	96.58	viprbrc.org
			RVA/Cow-wt/UGA/BUW-14-A035/2014/G12P [8] *	100	99.73	KX655531
			RVA/Human-wt/ESP/SS62505693/2013/G12P [8]	100	99.55	MH171356
			RVA/Human-wt/ESP/VI4816626/2012/G12P [8]	100	99.46	MH171365
VP6	I1	6	RVA/Human-wt/BGD/Matlab13/2003/G12P6	91.97	96.59	viprbrc.org
			RVA/Cow-wt/UGA/BUW-14-A035/2014/G12P [8] *	100	99.72	KX655532
			RVA/Human-wt/ESP/VI4816626/2012/G12P [8]	100	99.63	MH171381
			RVA/Human-wt/USA/VU12–13-149/2013/G12P [8]	100	99.63	KT919044
			RVA/Human-tc/USA/Wa-20-AG/1974/G1P [8] ***	100	99.9	KU861394
NSP3	T1	7	RVA/Human-wt/BEL/B3458/2003/G9P8	92.61	98.17	viprbrc.org
			RVA/Human-wt/ESP/SS62505693/2013/G12P [8]	100	99.5	MH171436
			RVA/Human-wt/ESP/VI4816626/2012/G12P [8]	100	99.4	MH171445
			RVA/Human-wt/ESP/SS257451/2012/G12P [8]	100	99.4	MH171434
			RVA/Human-tc/USA/Wa-20-AG/1974/G1P [8] ***	100	99.9	KU861398
NSP2	N1	8	RVA/Human-wt/BGD/Dhaka16/2003/G1P8	93.62	98.85	viprbrc.org
			RVA/Cow-wt/UGA/BUW-14-A035/2014/G12P [8] *	100	99.51	KX655535
			RVA/Human-wt/ESP/SS257451/2012/G12P [8]	100	99.41	MH171418
			RVA/Human-wt/ITA/PA99/12/2012/G12P8	100	99.41	KU048682
			RVA/Human-tc/USA/Wa-20-AG/1974/G1P [8] ***	100	99.9	KU861397
VP7	G3	9	RVA/Human-wt/BEL/B4106/2000/G3P14	97.74	85.19	viprbrc.org
			RVA/Human-wt/BRA/IAL-R594/2015/G3P [8] **	100	99.69	MH569775
			RVA/Human-wt/BRA/IAL-R330/2015/G3P [8] **	100	99.69	MH569773
			RVA Human-wt/JPN/Tokyo17–09/2017/G3P [8] **	100	99.59	LC477356
NSP4	E1	10	RVA/Human-wt/BGD/Dhaka6/2001/G11P25	75.32	98.48	viprbrc.org
			RVA/Cow-wt/UGA/BUW-14-A035/2014/G12P [8] *	100	99.86	KX655537
			RVA/Human-wt/ESP/SS257451/2012/G12P [8]	100	99.72	MH171450
			RVA/Human-wt/ITA/PA93/12/2012/G12P8	100	99.57	KU048725
			RVA/Human-tc/USA/Wa-20-AG/1974/G1P [8] ***	100	99.8	KU861399

* Cow-wt RVA ** Equine-Like Strains. *** Prototype Wa-like.

## Data Availability

Sequences were deposited with GenBank accession numbers MZ209037 to MZ209044.

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
