# Peer review of "Wa-1 Equine-Like G3P[8] Rotavirus from a Child with Diarrhea in Colombia"

_viruses, 2021, doi:10.3390/v13061075_

Round 1

Reviewer 1 Report

The authors have conducted full genome sequencing of one rotavirus strain from Colombia. It shows equine-like G3, P[8] strain close to both strains G12P[8] and G3P[8], with a Wa-like backbone. The strain is interesting and seemingly not published before in details. However, the manuscript needs improvement, so that the reader can understand the essential points. Language needs checking. Some conclusions need to be reconsidered. Below, specific comments are given for further improvement of the manuscript.

L 12 (Line 12) complete ’five’ (years?)

14 add ’and’ between six and five

17 check ’carried’ (carry?)

21 check ’focus’ (in focus?)

42 complete ’five’ (years?)

46 ’a member’

82 Should the referred article be number 5? Matthijnssens et al. 2011 paper is published before 2015.

87 ’suspensions’

101-2 complete ’…5’- and 3’-end…’

103 The supplementary table wasn't available for reviewing.

111 Was Sanger sequencing performed or was some other sequencing platform used? Please, give the GenBank accession numbers.

Table 1. It would be helpful if the Wa-strain, that seems to be close to the studied strain in many phylogenetic trees, would be included in Table 1.

143 ’displays’

How distantly (nucleic acid identity) related is the studied G3 sequence to other Wa-like G3 sequences (not equine like)?

Is VP4 tree representative for P[8] sequence clusters? Are other trees representatives for their sequences (meaning the clusters with letter codes, for example I1, I2 etc.). With which criteria were the sequences included in the tree selected?

Do the trees contain all essential prototype strains?

156 ’comparison’

Why do the authors not mention in the text the close identity with WA-strain in all the trees except G3 and possibly?

Based on phylogenetic tree, VP4 does not look exactly equine-like. This interpretation could be reconsidered.

174 Check ’an strong’

187 The value of a low nucleotide identity could be added here for the reader to assess.

201 Check ’this results’

202-4 This discussion seems not quite in place in this manuscript, since the strain was not DS1-like. Please, consider modifying the text. It could be discussed on how the new strain has evolved. Do you think this strain is the prevalent strain in Colombia?

211 ’lower primer specificity’

The vaccine used in Colombia could be added in the text.

236 ’in Brazil’

Author Response

The authors have conducted full genome sequencing of one rotavirus strain from Colombia. It shows equine-like G3, P[8] strain close to both strains G12P[8] and G3P[8], with a Wa-like backbone. The strain is interesting and seemingly not published before in details. However, the manuscript needs improvement, so that the reader can understand the essential points. Language needs checking. Some conclusions need to be reconsidered. Below, specific comments are given for further improvement of the manuscript.

R/. We agree to the reviewer and thank the helpful comments and corrections.

L 12 (Line 12) complete ’five’ (years?) -R/. Done

14 add ’and’ between six and five - R/. Done

17 check ’carried’ (carry?) - R/. Done

21 check ’focus’ (in focus?) -R/. Done

42 complete ’five’ (years?) -R/. Done

46 ’a member’ -R/. Done

82 Should the referred article be number 5? Matthijnssens et al. 2011 paper is published before 2015. - R/. Done

87 ’suspensions’- R/. Done

101-2 complete ’…5’- and 3’-end…’ -R/. Done

103 The supplementary table wasn't available for reviewing.

R/. We apologize for this unvoluntary mistake at the uploading process.

111 Was Sanger sequencing performed or was some other sequencing platform used? Please, give the GenBank accession numbers.

R/. We use SANGER sequencing. The sentence was edited. GenBank Accession numbers were included.

Table 1. It would be helpful if the Wa-strain, that seems to be close to the studied strain in many phylogenetic trees, would be included in Table 1.

143 ’displays’: -R/. Done

How distantly (nucleic acid identity) related is the studied G3 sequence to other Wa-like G3 sequences (not equine like)?

R/. The nucleic acid identity is low 55,4% to the Wa-like RVA/Human-tc/USA/Wa/1974/G1P8 (JX406755)

Is VP4 tree representative for P[8] sequence clusters? Are other trees representatives for their sequences (meaning the clusters with letter codes, for example I1, I2 etc.). With which criteria were the sequences included in the tree selected?

R/. Unfortunately, we did not prepare the tree for Subgenotype classification. Since we only have one sequence, we do not consider it pertinent to subclassify the Subgenotypes of P8

Do the trees contain all essential prototype strains?

R/. The trees included some of the prototype sequences. Most of the phylogenetic trees were built to highlight the origin of the sample and the difference between equine-like and Wa-like strains

156 ’comparison’. -R/. Done

Why do the authors not mention in the text the close identity with WA-strain in all the trees except G3 and possibly?

R/. We agree to the reviewer. It was only mentioned to highlight the wa-like structure of the Backbone. We modified the text to included this information

Based on phylogenetic tree, VP4 does not look exactly equine-like. This interpretation could be reconsidered.

R/. The equine-like classification is based on G3 not on P8. We modify the text to highlight this item.

174 Check ’an strong’ R/. Done

187 The value of a low nucleotide identity could be added here for the reader to assess.

R/. We agree to the reviewer. The information were included.

201 Check ’this results’ R/. Done

202-4 This discussion seems not quite in place in this manuscript, since the strain was not DS1-like. Please, consider modifying the text. It could be discussed on how the new strain has evolved. Do you think this strain is the prevalent strain in Colombia?

… R/. The text was modified according to the recommendation.

211 ’lower primer specificity’ R/. Done

The vaccine used in Colombia could be added in the text. R/. Done. Included in line 226

236 ’in Brazil’ R/. Done

Reviewer 2 Report

Everything is fine! Congratulations!

Author Response

Comments and Suggestions for Authors

Everything is fine! Congratulations!

R/. We thank the reviewer for him/her words. We improved the manuscript with all reviewers corrections, comments and suggestions.

Reviewer 3 Report

Retrospective near-full genome analysis of first Wa-1 equine-like G3P[8] rotavirus A strain in Columbia

By Marlen Martinez-Gutierrez et al (Corresponding author: Julian Ruiz-Saenz)

Submitted to Viruses(Editorial No. viruses-1210483)

General Comments

This is a ‘near-full genome’ analysis of a G3P[8] rotavirus A (RVA) strain isolated in Columbia, 2015, from a 2 y old child with diarrhea. The virus turned out to be an equine-like G3P[8] virus with a largely Wa-like genetic background. Regrettably, sequences of the genes encoding NSP1 (A) and NSP5 (H), thus of 2 out of 11 genes, are missing. Since equine-like RVA strains have been described before, an incomplete further sequence of an RVA of this constellation is not very satisfying. Supplementary Table 1 is also missing. It is suggested to complete the analysis of the genome of the Columbian RVA isolate. Clarification of several relatively minor points is also requested.

Specific Comments

Line

1          Reconsider Title, e.g. ‘Wa-equine-like G3P[8] virus isolated from a child with diarrhea in Columbia, 2015’, or similar.  

14        Read: … for six structural and 5-6 non-structural  proteins. …

17        … Here we carried out…

21        … Further studies must focus on…

24        Omit ‘as it has been reported worldwide’.

36        In 2016 the annual estimate of RVA-associated childhood deaths has furterh decreased to 128500. See: Troeger C, et al. Rotavirus Vaccination and the Global Burden of Rotavirus Diarrhea Among Children Younger Than 5 Years. JAMA Pediatr. 2018 Oct 1;172(10):958-965.

43        … Recent data showed that the prevalence of rotavirus infections had decreased…

46        … Reoviridae… (type in italics)

52        … RVA genomes have been… [similarly in line 56 by analogy]

64        … a high prevalence has been reported of…

70        Consider citation of: Martella V, et al. Zoonotic aspects of rotaviruses. Vet Microbiol. 2010 Jan 27;140(3-4):246-55.

81        Identify the RVA isolate analysed as done in Fig. 1.

82        Ref. [6] appears to be incorrect in the context.

103      Suppl. Table 1 is missing from the manuscript.

111      Details of the sequencing procedure applied should be reported.

114      The lack of NSP1 and NSP5 encoding sequences is not mentioned.

121      … compared with cDNA sequences obtained…

137      Read: … G12P[8]).

138ff   Table 1. Rephrase heading. The comment on line 114 also applies.

161      Fig. 2 could be transferred to Suppl. Mat.

168      Consider reading: … Here we describe the presence of the emerging…

186      … described by Akane et al that in comparison with… the Colombian VP7 gene had…

194      to 198. The sentence needs to be clarified by rephrasing.

201      … This result not only confirms…but also emphasizes the need of…

204      … with further evidence of reassortment of …

213      … The immune pressure associated with RVA vaccines has been…

236      … Brazil…

239      to 241. Omit sentence.

245      … due to the bovine gene sequences sharing high identities…

250      to 255. Consider omitting this paragraph.

Author Response

General Comments

This is a ‘near-full genome’ analysis of a G3P[8] rotavirus A (RVA) strain isolated in Columbia, 2015, from a 2 y old child with diarrhea. The virus turned out to be an equine-like G3P[8] virus with a largely Wa-like genetic background. Regrettably, sequences of the genes encoding NSP1 (A) and NSP5 (H), thus of 2 out of 11 genes, are missing. Since equine-like RVA strains have been described before, an incomplete further sequence of an RVA of this constellation is not very satisfying.

R/. We agree to the reviewer. However we were unable to amplify the genes encoding NSP1 (A) and NSP5 (H) even when we modify the amplification technique and the primer sequence. That’s why we denote as near-full (9 of 11), however, we understand and modify the title according to the recommendation.

Supplementary Table 1 is also missing. It is suggested to complete the analysis of the genome of the Columbian RVA isolate. Clarification of several relatively minor points is also requested.

R/. We apologize for this unvoluntary mistake at the uploading process.

Specific Comments

Line

1          Reconsider Title, e.g. ‘Wa-equine-like G3P[8] virus isolated from a child with diarrhea in Columbia, 2015’, or similar.  

R/. Done. We modified the tittle according to the reviewer recommendation.

14        Read: … for six structural and 5-6 non-structural  proteins. …-R/. Done

17        … Here we carried out…-R/. Done

21        … Further studies must focus on… R/. Done

24        Omit ‘as it has been reported worldwide’. R/. Done

36        In 2016 the annual estimate of RVA-associated childhood deaths has furterh decreased to 128500. See: Troeger C, et al. Rotavirus Vaccination and the Global Burden of Rotavirus Diarrhea Among Children Younger Than 5 Years. JAMA Pediatr. 2018 Oct 1;172(10):958-965.

R/. The reference was added and the paragraph updated.

43       Recent data showed that the prevalence of rotavirus infections had decreased… R/. Done

46        … Reoviridae… (type in italics) R/. Done

52        … RVA genomes have been… [similarly in line 56 by analogy] R/. Done

64        … a high prevalence has been reported of… R/. Done

70        Consider citation of: Martella V, et al. Zoonotic aspects of rotaviruses. Vet Microbiol. 2010 Jan 27;140(3-4):246-55. … R/. Done

81        Identify the RVA isolate analysed as done in Fig. 1. R/. Done

82        Ref. [6] appears to be incorrect in the context. R/. Done

103      Suppl. Table 1 is missing from the manuscript.

R/. We apologize for this unvoluntary mistake at the uploading process.

111      Details of the sequencing procedure applied should be reported. R/. Done

114      The lack of NSP1 and NSP5 encoding sequences is not mentioned. R/. Done

121      … compared with cDNA sequences obtained… R/. Done

137      Read: … G12P[8]). R/. Done

138ff   Table 1. Rephrase heading. The comment on line 114 also applies. R/. Done

161      Fig. 2 could be transferred to Suppl. Mat. R/. We couldn´t move it due to other reviewer suggestions.

168      Consider reading: … Here we describe the presence of the emerging… R/. Done

186     described by Akane et al that in comparison with… the Colombian VP7 gene had… R/. Done

194      to 198. The sentence needs to be clarified by rephrasing. … R/. Done

201      … This result not only confirms…but also emphasizes the need of… R/. Done

204      … with further evidence of reassortment of … R/. Done

213      … The immune pressure associated with RVA vaccines has been… R/. Done

236      … Brazil… R/. Done

239      to 241. Omit sentence. … R/. Done

245      … due to the bovine gene sequences sharing high identities…… R/. Done

250      to 255. Consider omitting this paragraph. … R/. Done

Round 2

Reviewer 1 Report

The authors have responded to the comments and those can be accepted. However, some more comments were necessary to give after reading the revised manuscript. Below, the specific comments are given for further improvement of the manuscript.

L14-L17 This long sentence has still several grammatical errors. Please, check carefully and modify. Proper writing might be easier if the long sentence were divided into several sentences.

L22-23 The last sentence in the abstract might be improved by changing the order (first: this study demonstrated…, second: further studies should focus on…)

L56 remove ‘countries’, not necessary.

L114 replace ‘deposit’ by ‘deposited’. The accession numbers are given for 8 sequences only. Please, add the number of the ninth sequence or explain why there are only 8 sequences in the GeneBank.

L135 Correct the ‘most of the constellation’ (for example: most sequences in the constellation?).

L157 The sentence needs modification (for example ‘nucleic acid sequences coding for structural and non-structural proteins…)

L194 replace ‘to be’ with ‘is’

L197 replace ‘potential’ with ‘potentially’

L248 add ‘virus’: ‘due to the bovine virus gene sequences…’

Supplementary table. Please, give references for the primers. Give locations of the primer sequences. In addition, tidy the general outlook of the sequences in the table.

A general comment: Would it be possible that the strain has evolved so that only VP7 RNA segment from the equine strain was reassorted with G12P[8] background (already having the ‘correct’ VP4) ?

Author Response

REVIEWER 1

The authors have responded to the comments and those can be accepted. However, some more comments were necessary to give after reading the revised manuscript. Below, the specific comments are given for further improvement of the manuscript.

L14-L17 This long sentence has still several grammatical errors. Please, check carefully and modify. Proper writing might be easier if the long sentence were divided into several sentences.

R/. We agree to the reviewer and modify the sentences.

L22-23 The last sentence in the abstract might be improved by changing the order (first: this study demonstrated…, second: further studies should focus on…)

R/. We agree to the reviewer and modify the sentence.

L56 remove ‘countries’, not necessary. R/. We agree to the reviewer and modify the sentence.

L114 replace ‘deposit’ by ‘deposited’. The accession numbers are given for 8 sequences only. Please, add the number of the ninth sequence or explain why there are only 8 sequences in the GeneBank.

R/. We deposited NINE (9) sequences from MZ209037 - MZ209044 to GenBank as follow:

  1. BankIt2460657 NSP2_RVA/Human-wt/COL-BUC/55/2015/G3P8    MZ209037
  2. BankIt2460657 NSP3_RVA/Human-wt/COL-BUC/55/2015/G3P8    MZ209038
  3. BankIt2460657 NSP4_RVA/Human-wt/COL-BUC/55/2015/G3P8    MZ209039
  4. BankIt2460657 VP1_RVA/Human-wt/COL-BUC/55/2015/G3P8      MZ209040
  5. BankIt2460657 VP2_RVA/Human-wt/COL-BUC/55/2015/G3P8      MZ209041
  6. BankIt2460657 VP3_RVA/Human-wt/COL-BUC/55/2015/G3P8      MZ209042
  7. BankIt2460657 VP4_RVA/Human-wt/COL-BUC/55/2015/G3P8      MZ209043
  8. BankIt2460657 VP7_RVA/Human-wt/COL-BUC/55/2015/G3P8      MZ209045
  9. BankIt2460889 VP6_RVA/Human-wt/COL-BUC/55/2015/G3P8      MZ209044

L135 Correct the ‘most of the constellation’ (for example: most sequences in the constellation?).

R/. We modified according to recommendation.

L157 The sentence needs modification (for example ‘nucleic acid sequences coding for structural and non-structural proteins…)

R/. We modified according to recommendation.

L194 replace ‘to be’ with ‘is’ - R/. We modified according to recommendation.

L197 replace ‘potential’ with ‘potentially’ R/. We modified according to recommendation.

L248 add ‘virus’: ‘due to the bovine virus gene sequences…’ R/. We modified according to recommendation.

Supplementary table. Please, give references for the primers. Give locations of the primer sequences. In addition, tidy the general outlook of the sequences in the table.

R/. We modified according to recommendation.

A general comment: Would it be possible that the strain has evolved so that only VP7 RNA segment from the equine strain was reassorted with G12P[8] background (already having the ‘correct’ VP4) ?

R/. We totally agree to the reviewer. It is quite possible that “backbone” native strain could be a G12P[8]. This information was included in the results; however, we added a short sentence in the discussion to be more accurate (Lines 207-209).

Reviewer 3 Report

Wa-1 equine-like G3P[8] rotavirus isolated from a child with diarrhea in Colombia, 2015

By Marlen Martinez-Gutierrez et al (Corresponding author: Julian Ruiz-Saenz)

Submitted to Viruses (Editorial No. viruses-1210483R1)

General Comments

This is the revised version of a manuscript the original submission of which has been studied and commented upon by this reviewer. The authors have followed most of the suggestions/comments, and the manuscript has improved. However, they virtually discarded the comment that 2 out of the 11 genes (encoding NSP1 and NSP5) of this rotavirus (RV) isolate have not been sequenced. This is peculiar since primers for these genes have been produced (as reported in the now supplied Suppl Table 1). What were the problems? The statement in line 135 does not suffice. In the Discussion authors state that ‘… This highlights the importance of carrying out full/whole genome analysis in order to accomplish active surveillance…’ (line 208f). Since co-circulating species A RVs reassort frequently, as also noted for some of the analysed genes in Discussion, data on NSP1 and NSP5 genes are very important, with NSP1 acting as an inhibitor of innate immune responses and NSP5 being an essential protein for viroplasm formation. The NSP1 gene in particular is the most variable gene of RVAs, and the primers used may not have recognised the gene of this isolate. As concluded before, ‘an incomplete further sequence of an RVA of this constellation is not very satisfying’. The suggestion to complete the sequence analysis of the genome of this RVA isolate has not been but should be followed.

Author Response

REVIEWER 3

This is the revised version of a manuscript the original submission of which has been studied and commented upon by this reviewer. The authors have followed most of the suggestions/comments, and the manuscript has improved. However, they virtually discarded the comment that 2 out of the 11 genes (encoding NSP1 and NSP5) of this rotavirus (RV) isolate have not been sequenced. This is peculiar since primers for these genes have been produced (as reported in the now supplied Suppl Table 1). What were the problems? The statement in line 135 does not suffice.

In the Discussion authors state that ‘… This highlights the importance of carrying out full/whole genome analysis in order to accomplish active surveillance…’ (line 208f). Since co-circulating species A RVs reassort frequently, as also noted for some of the analysed genes in Discussion, data on NSP1 and NSP5 genes are very important, with NSP1 acting as an inhibitor of innate immune responses and NSP5 being an essential protein for viroplasm formation. The NSP1 gene in particular is the most variable gene of RVAs, and the primers used may not have recognised the gene of this isolate. As concluded before, ‘an incomplete further sequence of an RVA of this constellation is not very satisfying’. The suggestion to complete the sequence analysis of the genome of this RVA isolate has not been but should be followed.

R/. We strongly agree to the reviewer comments and apologize for the lack of a satisfactory answer to him/her previous recommendations. The lack of amplification and analysis of the NSP1 and NSP5 adds a strong bias to the analysis of the full genome constellation of RVA. However, we do not have more sample to perform more analyses. We have included a paragraph discussing this bias as a weakness of our paper This is also one of the reasons why, we modified the Title and presented the paper as a Short paper.

Besides the natural variation of the NSP1/NSP5 genes and lower primer Specificity, it is also possible that, since samples from the child were took in 2015, the quality of the RNA were not optimal, and some fragments could be broken/degraded affecting the amplification of those genes. A new paragraph discussing those topics was included (lines 210 to 226).

We hope this new and honest Discussion included in our paper will be acceptable to the reviewer.